# Association of *SLCO1B3* and *SLCO1B1* Polymorphisms with Methotrexate Efficacy and Toxicity in Saudi Rheumatoid Arthritis Patients

**DOI:** 10.3390/ph18071069

**Published:** 2025-07-20

**Authors:** Rania Magadmi, Ahlam M. Alharthi, Lina A. Alqurashi, Ibtisam M. Jali, Zeina W. Sharawi, Maha H. Jamal, Yasser Bawazir, Mohammad Mustafa, Sami M. Bahlas, Basma T. Jamal, Hassan Daghasi, Abdulrahman S. Altowairqi, Dalal Sameer Al Shaer

**Affiliations:** 1Department of Clinical Pharmacology, Faculty of Medicine, King Abdulaziz University, Jeddah 21589, Saudi Arabia; 2Department of Biological Sciences, Faculty of Science, King Abdulaziz University, P.O. Box 80203, Jeddah 21589, Saudi Arabia; 3Department of Medicine, Faculty of Medicine, King Abdulaziz University, Jeddah 21589, Saudi Arabia; 4Rheumatology Unite, Department of Medicine, Faculty of Medicine, University of Jeddah, Jeddah 21589, Saudi Arabia; 5King Fahad Medical Research Center, King Abdulaziz University, Jeddah 21589, Saudi Arabia; 6Division of Rheumatology, Internal Medicine Department, Al Hada Armed Forces Hospital, Taif 26792, Saudi Arabia; 7Department of Genetic Medicine, Faculty of Medicine, University of Jeddah, Jeddah 21589, Saudi Arabia

**Keywords:** methotrexate, personalized medicine, pharmacogenetics, polymorphism, rheumatoid arthritis, single nucleotide polymorphism

## Abstract

**Background:** Methotrexate (MTX) remains the most commonly prescribed drug used to treat rheumatoid arthritis (RA). Polymorphisms in solute carrier organic anion transporter family member 1B3 (*SLCO1B3*) and *SLCO1B1* may play a critical role in MTX pharmacokinetics and patient outcomes. However, research on these polymorphisms in Saudi Arabia remains limited. We evaluated the association of *SLCO1B3* (rs4149117, rs7311358) and *SLCO1B1* (rs2306283, rs4149056) polymorphisms with MTX efficacy and safety in Saudi patients with RA. **Methods:** This multicenter, case-control study included patients diagnosed with RA in Jeddah and Taif. Demographic and clinical data were collected and analyzed. Genotyping of *SLCO1B3* (rs4149117, rs7311358) and *SLCO1B1* (rs2306283, rs4149056) polymorphisms was performed using Sanger sequencing. Statistical analyses, including logistic regression and haplotype analysis, were conducted to evaluate associations between these polymorphisms, MTX efficacy, and toxicity. **Results:** The study cohort comprised 100 patients with RA, with 46 showing a good response to MTX and 54 showing a poor response. Clinical predictors of MTX response were significantly higher in patients with poor response. Both *SLCO1B3* polymorphisms (rs4149117, rs7311358) were significantly associated with anemia. Significant associations were found between *SLCO1B1* (rs2306283) and gastrointestinal disturbances and anemia. The GAAT haplotype was significantly more prevalent among good responders, while the TGGT haplotype was significantly associated with poor responders. **Conclusions:** These results highlight the importance of genetic testing in predicting MTX treatment outcomes and tailoring personalized treatment plans for patients with RA to improve efficacy and minimize adverse effects.

## 1. Introduction

Rheumatoid arthritis (RA) is a chronic, systemic, autoimmune inflammatory disorder that primarily affects synovial joints. It is characterized by joint pain, stiffness, and swelling, often progressing to joint destruction, disability, and reduced quality of life [1]. Globally, according to the World Health Organization (WHO), RA affects approximately 1% of the global population. Prevalence varies by region, ranging from 0.5 to 1% in the United States to as high as 5% in other regions [2]. The incidence of RA is approximately 29 cases per 100,000 individuals annually. In Saudi Arabia, prevalence is estimated between 0.4 and 0.5% [3,4].

Disease-modifying antirheumatic drugs (DMARDs) are a class of medications used to treat RA and other autoimmune diseases, mitigating inflammation and preventing joint damage by modulating immune system activity [5]. Methotrexate (MTX) is a widely used DMARD for RA; however, responses to MTX vary among individuals, partly owing to genetic polymorphisms [6]. MTX is transported to cells via specific influx transporters. Influx proteins are membrane transporters that facilitate the movement of substances across the cell membrane. The organic anion-transporting polypeptide (OATP) family is involved in the active transport of specific molecules or ions into cells, allowing them to enter the intracellular space [7].

Eleven members of the human OATP family are encoded by solute carrier organic anion transporter (*SLCO*) genes. OATP1B3 and OATP1B1, encoded by *SLCO1B3* and *SLCO1B*, respectively, are well-characterized and highly polymorphic [8]. *SLCO1B3* encodes the novel human polypeptide OATP1B3, an influx transporter protein found in the liver. OATP1B3 plays a crucial role in the hepatic uptake of medications such as statins (e.g., atorvastatin and pravastatin), anticancer drugs (e.g., irinotecan and docetaxel), and antibiotics (e.g., rifampin). MTX is a substrate for OATP1B3 transport [9].

*SLCO1B1* encodes a liver-specific member of OATP1B1. The encoded transmembrane receptor protein facilitates the sodium-independent uptake of various endogenous molecules, such as 17-beta-glucuronosyl estradiol, bilirubin, and leukotriene C4. It is primarily expressed in the basolateral membrane of hepatocytes and is essential in the uptake of various endogenous and exogenous anionic compounds, including MTX [9].

The major clearance pathway of MTX from the systemic circulation is through its transport into hepatocytes via *SLCO1B1*. OATP1B1 and OATP1B3, members of the OATP family transporters, are exclusively expressed in human hepatocytes, playing essential roles in the hepatic uptake and clearance of organic compounds [10].

Genetic variations in *SLCO1B3* and *SLCO1B1* affect the function of transporters, influencing MTX pharmacokinetics and efficacy [11]. Studies have investigated the association of *SLCO1B3* and *SLCO1B1* with MTX efficacy in patients with RA. Certain *SLCO1B3* and *SLCO1B1* genetic variants are associated with lower MTX efficacy, whereas others report no significant association [12]. Despite the numerous studies on *SLCO* gene variants and MTX efficacy, comprehensive research specifically addressing both *SLCO1B3* and *SLCO1B1* variants in the context of MTX treatment in Saudi patients with RA is limited. This gap highlights the need for targeted research to optimize treatment strategies for this population. Therefore, this study aimed to investigate the correlation of genetic polymorphisms of *SLCO1B3* and *SLCO1B1* to the efficacy and safety of MTX in patients with RA in Saudi Arabia. The results of this study will provide a better understanding of RA pathophysiology, potentially identifying new therapeutic strategies.

## 2. Results

### 2.1. Demographic and Clinical Data

This study included 100 patients with RA. As shown in Table 1, the mean age of the patients was 53.02 years (SD = 12.78) and the mean age at onset was 46.26 years (SD = 12.35). Most participants were female (80%) and had been diagnosed with RA for more than a year. Most patients (87%) had been diagnosed with RA for more than a year.

The average MTX dose was 14.98 mg (SD = 3.7), and 85% of patients were on MTX treatment for more than a year. Regarding disease activity, the mean DAS-28 score was 2.94 (SD = 0.96), CRP level was 10.56 mg/dL (SD = 7.40), and the erythrocyte sedimentation rate (ESR) was 27.44 mm/h (SD = 17.33). Rheumatoid factor (RF) was positive in 68% of patients, whereas anti-CCP was positive in 62%. The most common adverse reaction was GI upset (61%). Increased liver enzymes were reported by 25% of participants, while anemia was reported in 14%.

Regarding MTX treatment response, 46% of patients were classified as good responders and 54% were classified as poor responders.

Table 2 presents demographic and clinical factors associated with MTX response. No significant differences (*p* > 0.05) were observed in age, age at onset, sex, nationality, or MTX dose between good responders (N = 46) and poor responders (N = 54). However, a significant difference was observed in disease duration, with a longer duration associated with a poor response (*p* = 0.009). Among poor responders, 96.3% had a disease duration >1 year, compared to 76.1% of good responders.

Steroids use was significantly higher among the poor responders (74.1% vs. 34.6%, *p* < 0.001). As expected, disease activity markers DAS-28, CRP, and ESR were significantly higher in poor responders (*p <* 0.001, *p <* 0.001, and *p =* 0.046, respectively). No significant differences were observed in RF and anti-CCP positivity between the two groups. Furthermore, the incidence of adverse drug reactions was similar in both groups (Table 2).

### 2.2. Genetic Association Data for SLCO1B3 Polymorphisms

#### Genotype Distributions and Allele Frequencies of SLCO1B3 Polymorphisms

Table 3 provides a summary of genotype distributions and allele frequencies of two selected *SLCO1B3* SNPs: rs4149117 located on exon 4 and rs7311358 on exon 7. Notably, genotype distributions of both tested SNPs did not align with Hardy–Weinberg equilibrium. Both polymorphisms significantly deviated from the expected equilibrium proportions (*p* < 0.001 * for rs4149117 and rs7311358).

For *SLCO1B3* (rs4149117), the TT genotype was observed in 26% of samples, whereas the GG genotype was more prevalent (74%). In *SLCO1B3* (rs7311358), the AA genotype was predominant at 81%, with AG and GG genotypes at 11% and 8%, respectively.

Next, the genotype distributions in this study were compared with those of other populations, using data from the National Center for Biotechnology Information (NCBI) website (https://www.ncbi.nlm.nih.gov/, accessed on 17 July 2025). The results indicated ethnic variations among the different populations, as shown in Figure 1.

In particular, the SNP (rs4149117) genotype distribution in this cohort was comparable to that in Middle Eastern and American populations, as illustrated in Figure 1A. Notably, the prevalence of the GG genotype was higher in all populations, except in the African population. In contrast, the study cohort showed no TG genotypes, whereas other populations exhibited varying TG genotype frequencies. The rs7311358 genotype distribution showed a higher prevalence of AA genotypes in the study population. This trend is similar to that observed in the Middle Eastern and American populations (Figure 1B).

### 2.3. Association Between SLCO1B3 Gene Polymorphisms and Patient Response to MTX

The association between the two *SLCO1B3* gene polymorphisms (rs4149117 and rs7311358) and MTX treatment response was evaluated by comparing genotype and allele frequencies between good and poor responders (Table 4). For the rs4149117 polymorphism, TT and GG genotype distribution did not significantly differ between good (23.9% TT, 76.1% GG) and poor responders (27.8% TT, 72.2% GG). T and G allele frequencies showed no significant differences between the two groups. Similarly, for the rs7311358 polymorphism, no significant differences were observed in genotype distribution (AA, AG, and GG) or allele frequencies (A and G) between good and poor responders.

### 2.4. Association Between the Haplotypes of SLCO1B3 Polymorphisms and Their Response to MTX

Table 5 shows haplotype frequencies of *SLCO1B3* polymorphisms (rs4149117 and rs7311358) in good and poor responders to MTX treatment. D0 values for rs4149117 and rs7311358 were 0.79 and 0.35, respectively, with *p*-values > 0.001. The global haplotype association *p*-value was 0.67.

Four haplotypes were identified in the study population (Table 5), with the GA haplotype being the most common in both groups (70 good responders and 78 poor responders). However, no significant differences in MTX response were observed based on haplotype frequencies of SLCO1B3 polymorphisms.

### 2.5. Associations Between SLCO1B3 Gene Polymorphisms and the Risk of MTX Adverse Reactions

No significant associations were found between *SLCO1B3*SNPs (rs4149117 and rs7311358) genotype distributions and the risk of GI disturbance or increased liver enzyme levels. Notably, both polymorphisms showed a significant correlation with anemia (Table 6).

### 2.6. Genetic Association Data for SLCO1B1 Polymorphisms

#### Genotype Distributions and Allele Frequencies of SLCO1B1 Polymorphisms

Table 7 provides a summary of the genotype distributions and allele frequencies for *SLCO1B1* SNPs rs2306283 and rs4149056, both located on exon 5. Notably, genotype distributions of both SNPs aligned with Hardy–Weinberg equilibrium (*p =* 0.8, 0.6, for *SLCO1B1* polymorphisms rs2306283 and rs4149056, respectively). Moreover, no significant differences were observed across any subgroups (All *p* > 0.05).

For the *SLCO1B1* (rs2306283) polymorphism, the AA genotype was observed in 17% of samples, GG in 33%, and AG was more prevalent at 50%. In the *SLCO1B1* rs4149056 polymorphism, the TT genotype was predominant in 52% of samples, with CC and TC genotypes at 9% and 39%, respectively.

Next, the distribution of the *SLCO1B1* genetic variants rs2306283 and rs4149056 observed in this study was compared with NCBI data (https://www.ncbi.nlm.nih.gov/, https://www.ncbi.nlm.nih.gov/, accessed on 17 July 2025), showing ethnic variation among the populations (Figure 2). For rs2306283 (Figure 2A), the GG genotype was more prevalent in other populations than that in the study population. Conversely, the study population had a higher frequency of the AG genotype than other populations, exhibiting varying frequencies of the AG genotype.

In rs4149056 (Figure 2B), the TT genotype was most prevalent across all populations, including the study population. The genotype distribution for the study population was similar to that of Middle Eastern and American populations.

### 2.7. Association Between SLCO1B1 Gene Polymorphisms and Patient Response to MTX

To evaluate the association between the two *SLCO1B1* gene polymorphisms (rs2306283 and rs4149056) and response to MTX treatment, their genotype and allele frequencies were compared between good and poor responders (Table 8). For the rs2306283 polymorphism, the distribution of the AA, AG, and GG genotypes did not significantly differ between good and poor responders. The allele frequencies (A and G) were not significantly different between the two groups. Similarly, for the rs4149056 polymorphism, no significant differences were observed in genotype distribution (TT, TC, and CC) or allele frequencies (T and C) between good and poor responders.

### 2.8. Association Between SLCO1B1 Polymorphism Haplotypes and Response to MTX

The results presented in Table 9 show the haplotype frequencies of *SLCO1B1* polymorphisms (rs2306283 and rs4149056) in good and poor responders to MTX treatment. The linkage disequilibrium (D0) values for rs4149117 and rs7311358 were 0.122 and 0.67, respectively, with *p* > 0.001. The global haplotype association *p*-value was 0.67.

Four haplotypes were identified in the study population (Table 9). Significant differences were observed in patient responses to the MTX haplotype frequencies of *SLCO1B1* polymorphisms. Good responders had an AT haplotype combination compared to poor responders. Logistic regression showed poor responders had 0.46 times lower odds of having the AT haplotype than good responders.

### 2.9. Associations Between SLCO1B1 Gene Polymorphisms and the Risk of MTX Adverse Reactions

Genotype distribution of the *SLCO1B1* rs2306283 SNP was significantly associated with GI disturbances and anemia; however, it was not associated with increased liver enzyme levels among patients with RA receiving MTX treatment (Table 10).

In contrast, the *SLCO1B1* rs4149056 SNP was significantly associated with GI disturbance and increased liver enzyme levels. No association was found between rs4149056 and anemia in this cohort.

### 2.10. Association Between SLCO1B3 and SLCO1B1 Polymorphism Haplotypes and the Response to MTX

Significant differences in the haplotype frequencies of the *SLCO1B3* and *SLCO1B1* polymorphisms between good and poor responders to MTX treatment were observed (Table 11). Good responders had a significantly higher frequency of the GAAT haplotype than poor responders. Conversely, poor responders had a significantly higher frequency of the TGGT haplotype than good responders. Notably, the GGGT haplotype was observed only in the good responder group, whereas the TGAC and GGAT haplotypes were found exclusively in the poor responder group.

## 3. Discussion

RA is a systemic autoimmune disease characterized by inflammatory arthritis and extraarticular manifestations. If left untreated, RA can progress, increasing morbidity and mortality rates. MTX is considered the first-line DMARD, although its efficacy varies. Approximately 30–45% of patients experience adverse reactions to MTX therapy, leading to discontinuation in 16% of cases. Genetic factors significantly contribute to this variability, with pharmacogenetics increasingly used to explore individual genetic variations that influence treatment outcomes [13]. Therefore, this study aimed to investigate the correlation between SLCO1B3 and SLCO1B1 genetic polymorphisms with MTX efficacy and safety in Saudi Arabia. Several factors can affect patient response to MTX; thus, in addition, this study aimed to identify patients likely to respond well to MTX based on demographic, clinical, and genetic factors. Early prediction of a patient’s reaction to MTX could save costs, time, and improve health outcomes.

Demographic analysis revealed an average RA onset of 46.26 years, aligning with the literature indicating that RA frequently begins in the middle-aged. Notably, the results showed a predominance of female participants (80%), consistent with well-documented associations between RA and female sex, likely owing to hormonal influences and specific genetic risk factors [14,15]. Research has indicated that hormonal factors, particularly estrogen, play a role in RA development and progression owing to their immunomodulatory effects, and may influence autoimmune responses. Hormonal changes during different life stages such as puberty, pregnancy, and menopause can impact disease activity in women with RA. Moreover, specific genetic variants, such as the HLA-DRB1 and shared epitope alleles, are strongly associated with RA risk in women [14,15]. Although RA is more common in females, it can still occur in males, albeit less frequently.

The primary findings of this study examined the response characteristics of 100 patients with RA treated with MTX. The results showed no difference in age, sex, or MTX dosage between good and poor responders, consistent with previous studies conducted by Duong et al. [16] and Majorczyk et al. [17]. Patient age did not correlate with treatment response to MTX, as both younger and older individuals were found to either respond well or poorly to therapy. Although sex differences exist in RA prevalence and severity, MTX response was not significantly influenced by sex, with both male and females responding similarly.

Notably, patients with longer disease duration were more susceptible to a poor response to MTX. A significant association was observed between disease duration and MTX response, with 96.3% of poor responders having a longer disease duration than 76.1% of good responders. This finding supports previous research showing that longer RA duration is associated with poorer MTX response [16]. Several factors may have contributed to this observation, the first being disease progression, were increased joint inflammation and structural damage make disease management more difficult for MTX or other DMARDs. Second, accumulated damage from uncontrolled inflammation can cause irreversible joint changes such as deformities and erosions, limiting treatment responses. Third, autoimmune changes in long-standing RA may lead to more complex and heterogeneous autoimmune responses, increasing resistance to specific treatments, including MTX. Fourth, comorbidities and medications over time may also influence MTX response and reduce its efficacy. Fifth, long-term use of MTX in RA may result in altered pharmacokinetics, potentially reducing its effectiveness over time [18], with research findings indicating that MTX monotherapy may decrease in efficacy and patient compliance over time.

Another demographic predictor of MTX responsiveness in the current study was steroid use, which was significantly higher in the poor responders. Specifically, steroid use was significantly higher in poor responders (74.1% vs. 34.6%, *p <* 0.001). Existing research has shown that poor responders to MTX often have more severe disease activity, including higher inflammation, joint damage, and functional impairment [16]. In such cases, healthcare providers may prescribe steroids to control acute symptoms and manage disease flares, especially in the early stages of treatment or when the full therapeutic effects of MTX have not yet been achieved. Moreover, steroids may serve as a “bridge” therapy, providing symptomatic relief, while waiting for MTX to achieve its full therapeutic effect, which typically takes several weeks to months. In addition, in cases where MTX monotherapy is insufficient to control disease activity, combination therapy with steroids may be considered for better disease control [19].

This study revealed that the markers of disease activity, including DAS-28, CRP, and ESR, were significantly higher in poor responders. In addition, a previous study showed higher DAS-28, CRP, and ESR levels in poor responders, with several contributing factors. First, poor responders to MTX often have ongoing disease activity and inflammation that are not adequately controlled by medication, resulting in high tender and swollen joint counts, severe symptoms, and systemic inflammation. Consequently, DAS-28, which incorporates joint counts and global patient assessments, may be higher. Second, although MTX is an anti-inflammatory medication, it may not suppress inflammation effectively in some patients or it may require more time to achieve optimal control. This can lead to higher CRP and ESR levels, markers of systemic inflammation. Third, poor responders to MTX often had more severe disease activity at baseline. These patients may present with more tender and swollen joints, significant functional impairments, contributing to higher DAS-28, CRP, and ESR levels. Fourth, MTX may require several weeks or months to achieve full therapeutic effect, during which disease activity and inflammation persisted in poor responders, leading to elevated scores. Fifth, additional factors such as comorbidities, including obesity or other inflammatory conditions, may contribute to higher DAS-28 scores, CRP, and ESR values in poor responders [17].

Notably, the DAS-28 score, CRP levels, and ESR are all tools used to assess disease activity and inflammation in RA. These measures provide objective data to evaluate treatment responses and guide therapeutic decisions. When these indicators are elevated in poor responders to MTX, further evaluation, potential treatment adjustments, or alternative therapies may be needed to achieve better disease control. In contrast, RF and anti-CCP antibodies are well-established diagnostic markers of RA. However, no significant differences were observed in the percentages of RF and anti-CCP positivity between good and poor responders to MTX. Similarly, previous studies have shown that RF and anti-CCP antibodies are not associated with response to MTX therapy in patients with established RA [20].

Collectively, the demographic and clinical findings of this study emphasized the importance of considering patient characteristics when selecting appropriate treatment options. This may also be attributed to the influence of pharmacogenetics, which examines how genetic factors influence an individual’s response to medication. Pharmacogenetics, a growing field of pharmacology, has gained considerable attention, owing to its potential to improve treatment outcomes [19]. By understanding how genes affect drug metabolism, efficacy, and adverse reactions, healthcare professionals can tailor medication dosage and treatment duration more effectively. This personalized approach can lead to improved treatment plans and better patient outcomes.

Increasing evidence indicates that genetic factors have a substantial impact on individual differences in response to MTX. Recently, the influence of genetic factors on the pharmacokinetics and pharmacodynamics of MTX has been a research focus, with specific focus on the role of SNPs in altering these aspects [19]. Acknowledging the involvement of pharmacogenetics in MTX response is crucial for the future of personalized treatment of patients with RA. This variability may be a consequence of MTX pharmacokinetic changes, partly owing to SNPs in genes encoding MTX membrane transporter proteins, affecting influx and/or efflux [21]. MTX is transported to cells via specific influx transporters. Influx proteins are membrane transport proteins that play crucial roles in the movement of various substances across the cell membrane. These proteins are involved in the active transport of specific molecules or ions into cells, allowing them to enter the intracellular space [7]. MTX transporters are expressed in several tissues and affect the absorption, distribution, and/or elimination of the drug. In the GI tract, MTX is absorbed at the apical membrane of enterocytes via solute carrier (SLC) family 19 member 1 (*SLC19A1*)-mediated active transport and probably SLC family 46 member 1 (*SLC46A1*). SLC19A1, SLCO family member 1B3, and SLC organic anion transporter family member 1B1 (*SLCO1B1*) are responsible for the hepatic uptake of MTX [21].

*SLCO1B3* encodes a novel human organic anion-transporting polypeptide 1B3 (OATP1B3), an influx transporter protein found in the liver. OATP1B3 plays a crucial role in the hepatic uptake of numerous medications, including statins (e.g., atorvastatin and pravastatin), anticancer drugs (e.g., irinotecan and docetaxel), and certain antibiotics (e.g., rifampin). MTX is a substrate for OATP1B3 transport [9].

*SLCO1B1* encodes a liver-specific member of OATP family (OATP1B1). This protein is a transmembrane receptor that mediates the sodium-independent uptake of various endogenous compounds, including bilirubin, 17-beta-glucuronosyl estradiol, and leukotriene C4. It is primarily expressed in the basolateral membrane of hepatocytes and is vital for the uptake of various endogenous and exogenous anionic compounds, including MTX. Hepatic uptake of MTX via *SLCO1B1* is the primary pathway for MTX clearance from systemic circulation [10].

As MTX is transported by both *SLCO1B3* and *SLCO1B1*, and no studies that have analyzed the contribution of SNPs in *SLCO1B3* to MTX in RA are available, conducting combined studies to address the importance of these SNPs is necessary. Studying genetic variations in *SLCO1B3* and *SLCO1B1* could offer valuable insights into personalizing treatment strategies. Exploring genetic factors associated with drug metabolism and transport may help us understand why some patients experience adverse drug reactions or fail to respond to treatment [22].

In this study, the most commonly reported SNPs of *SLCO1B3* (rs4149117 and rs7311358) and *SLCO1B1* (rs2306283 and rs2306283) have been studied for their potential association with MTX responsiveness and safety. We provided a comprehensive analysis of genetic variations within *SLCO1B3* and their potential impact on MTX treatment response in patients with RA. The significant deviation from Hardy–Weinberg equilibrium observed for SNPs rs4149117 and rs7311358 may be attributed to the sample population consisting of patients with RA rather than healthy individuals. This deviation showed a potential population structure. Such deviations highlight the importance of considering disease-specific genetic backgrounds when interpreting pharmacogenetic data.

The *SLCO1B3* polymorphism genotype distribution among our study population revealed a higher prevalence of the GG genotype for rs4149117 and the AA genotype for rs7311358, consistent with findings from other populations, including those in the Middle East (NCBI, 2022). This indicated a possible ethnic or geographical influence on SNP distribution, supporting the findings of Suarez-Kurtz et al. (2012), who reported similar geographical disparities in *SLCO1B3* polymorphisms [23].

No significant association was observed between the genotypes of *SLCO1B3* (rs4149117) and (rs7311358) and the response to MTX treatment in patients with RA. This is aligned with the results reported by Banach et al. [24], who found no significant influence of these *SLCO1B3* polymorphisms on MTX efficacy in a cohort of 151 pediatric patients with acute lymphoblastic leukemia (ALL). However, prior studies investigating the impact of *SLCO1B3* polymorphisms on MTX response in patients with RA are lacking, highlighting a gap in the current literature.

In pharmacogenetics, haplotype analysis of specific genetic variants is a powerful tool for identifying potential drug–response associations. It is particularly relevant when investigating *SLCO1B3* SNPs in patients who respond differently to MTX treatment. These analyses aim to identify patterns that may predict therapeutic outcomes or adverse reactions. Despite the theoretical promise of this method, our analysis of *SLCO1B3* haplotypes among good and poor responders to MTX revealed no significant association with treatment response. This highlighted the complexity of drug response genetics, influenced by multiple interacting factors beyond single-gene haplotypes. Notably, larger and more diverse study populations are needed to validate these findings and potentially reveal subtle genetic effects that smaller cohorts may not.

Exploring genetic polymorphisms associated with adverse drug reactions is crucial for personalized medicine, particularly for chronic conditions such as RA. MTX is a cornerstone treatment for RA; however, it is known for causing considerable side effects, including GI disturbances, liver enzyme elevation, and anemia [13]. Therefore, this study focused on determining whether specific polymorphisms in *SLCO1B3* are associated with an increased risk of adverse drug reactions. Analysis of *SLCO1B3* (rs4149117) and *SLCO1B3* (rs7311358) polymorphisms revealed no statistically significant association with GI disturbances or increased liver enzyme levels. This indicates that these specific *SLCO1B3* polymorphisms may not be useful predictors of GI and liver side effects in patients treated with MTX. Notably, although the aforementioned adverse effects showed no association, the *SLCO1B3* (rs4149117) and *SLCO1B3* (rs7311358) polymorphisms were significantly associated with anemia. This indicates a potential genetic influence on MTX-related hematological toxicity, requiring further investigation. This observation is particularly noteworthy, as it may indicate how *SLCO1B3* affects MTX metabolism or transport in a manner that affects bone marrow function or erythropoiesis.

Overall, the absence of significant differences in genotype distributions between good and poor responders to MTX showed that these *SLCO1B3* polymorphisms may not be reliable markers for predicting MTX responses. This may be attributed to the complex nature of MTX transport and metabolism, involving multiple genes and pathways [25].

Although the results concerning *SLCO1B3* polymorphisms in this study were not significant, analyzing *SLCO1B1* polymorphisms further elucidated a broader spectrum of transporter gene variations that collectively influence pharmacokinetic and pharmacodynamic outcomes in clinical settings. Both genes encode organic anion-transporting polypeptides that significantly influence the pharmacokinetics of various drugs, including MTX. Polymorphic variants within these genes can alter transporter expression and function, affecting drug absorption, distribution, metabolism, and excretion (ADME) [26]. Thus, understanding genetic variations in *SLCO1B1* complements the findings from *SLCO1B3* and enriches the comprehension of the broader genetic factors influencing drug efficacy and safety. This integrated view is pivotal for advancing personalized medicine and emphasizes the need for comprehensive genetic screening to optimize therapeutic strategies across different patient populations.

Two well-characterized functional *SLCO1B1* SNP variants, *SLCO1B1* (c.388A  >  G rs2306283) and *SLCO1B1* (c.521 T  >  C rs4149056), were studied in this cohort. The genotype distributions for both SNPs were in Hardy–Weinberg equilibrium, which is essential for population genetic studies, indicating a lack of evolutionary pressure or non-random mating in the sample population. Moreover, the genotype distributions from this cohort were compared with the global data available in the NCBI database, revealing significant ethnic variations (NCBI, 2022). For rs2306283, this study population exhibited a higher frequency of the AG genotype than other populations, which may explain differences in drug responses across ethnic groups.

The genotype distribution and allele frequencies in this study revealed no significant association between *SLCO1B1* (rs4149117 and rs7311358) and the response to MTX treatment in patients with RA, contrasting with earlier findings. Jenko et al. [27] showed that *SLCO1B1* rs2306283 was associated with a higher DAS-28 score after 6 months of MTX monotherapy in Slovenian patients with RA. Similarly, a previous study of 322 Chinese children treated with MTX for ALL showed that children with the rs4149056 CC genotype exhibited poorer outcomes than those with the TT or TC genotypes [28]. Notably, these findings in patients with ALL may not universally apply to other MTX-sensitive diseases, demonstrated by the lack of a significant association between *SLCO1B1* SNPs and MTX clearance in patients with osteosarcoma [29].

In contrast, the haplotype frequencies of *SLCO1B1* for rs4149117 and rs7311358 were significantly different in patient responses to MTX. Good responders had the AT haplotype combination in poor responders, indicating that it may be a beneficial factor for MTX. Logistic regression supported this finding, showing that the odds of being a poor responder were significantly lower in individuals with the AT haplotype, with an OR of 0.46. This indicates that carriers of the AT haplotype have nearly half the risk of a poor response compared with non-carriers, which was statistically significant.

In addition to response variability, *SLCO1B1* polymorphisms appeared to influence the risk of MTX-induced adverse reactions. Genotype distributions of rs2306283 and rs4149056 were significantly associated with GI disturbances. This observation is consistent with findings from a previous study conducted on 233 Portuguese patients with RA receiving MTX, which also reported significant associations between TT and TC genotypes and GI disturbances [21]. Such consistent findings across different cohorts highlight the potential of pharmacogenetic insights in predicting adverse effects, tailoring individualized treatment plans for patients with RA undergoing MTX therapy.

Moreover, individuals with the AG genotype at rs2306283 exhibited significantly higher anemia rates. A similar result was reported in a previous study, indicating that the *SLCO1B1* rs2306283 GG genotype is associated with an increased risk of MTX-induced anemia [30]. This indicates that *SLCO1B1* may play a role in modulating MTX’s impact on erythropoiesis or bone marrow function. Furthermore, the TT genotype of rs4149056 was significantly associated with increased liver enzyme levels, whereas the CC genotype showed protective effects against hepatotoxicity. In contrast, a previous meta-analysis revealed that the *SLCO1B1* 521T > C polymorphism affected MTX clearance, and C allele carriers showed a 40–60% reduction in MTX clearance compared to TT wild-type homozygote carriers. Thus, the increased MTX toxicity in patients with the *SLCO1B1* CC genotype is partially attributable to decreased MTX clearance [6]. These results have also been reported in Chinese children with ALL [31]. Martinez et al. showed an opposing trend in MTX-induced hepatotoxicity among SLCO1B1 521T > C carriers, although it was not statistically significant. This inconsistent result was possibly due to ethnic differences between the study. The potential application of our findings has been emphasized, suggesting that pre-treatment genotyping of SLCO1B1 and SLCO1B3 could identify high-risk patients for closer monitoring and personalized treatment strategies to improve MTX efficacy and safety. No literature is presently available on the association between *SLCO1B1* SNPs and the development of MTX-induced liver toxicity among patients with RA.

The *SLCO1B1* 521T > C polymorphism is associated with drug-induced adverse events from statins and tegafur–uracil. To this end, the risk of statin-induced side effects, including abnormal alanine transaminase levels, has been reported in a previous meta-analysis. Further, patients with breast cancer with the CC or CT genotype receiving tegafur–uracil were reported to have a higher risk of elevated aspartate aminotransferase levels than those with the TT genotype. These findings substantiate the relationship between the *SLCO1B1* 521T > C polymorphism and drug-induced liver injury [32].

Genetic variations in *SLCO1B1* have been associated with different types of MTX toxicity. For example, the *SLCO1B1* rs2306283 variant was found to increase the risk of MTX-induced neurotoxicity in Brazilian pediatric patients with ALL [33]. Eldem et al. revealed that specific *SLCO1B1* variants (rs4149056 CC and rs11045879) are associated with poorer MTX tolerance in Turkish children undergoing maintenance therapy for ALL [43 p2028]. Additional research indicated significant differences in MTX elimination times linked to the rs2306283 SNP, where AA genotype carriers experienced delayed MTX elimination, increasing the risk of severe side effects such as oral mucositis, hepatotoxicity, and myelosuppression [34]. In patients with B-cell lymphoma, the rs11045879 CT genotype leads to higher MTX plasma concentrations, whereas the CC genotype increases neutropenia risk [35].

Previous studies have investigated associations between other *SLCO1B1* SNPs (rs11045879 and rs4149081) and MTX toxicities in various populations, including Chinese [36]. However, these studies did not find any significant associations, highlighting the potential variability in the effects of specific SNPs on drug toxicity across ethnic groups, emphasizing the importance of considering genetic diversity in pharmacogenetic research. Collectively, these findings are particularly relevant for clinical management as they indicate that genetic testing for *SLCO1B1* could be beneficial in predicting adverse effects, potentially guiding personalized MTX therapy.

The significant association of *SLCO1B1* with MTX response, compared to the lack of association with *SLCO1B3*, could be explained. *SLCO1B1* is highly expressed in hepatocytes and plays a critical role in mediating the hepatic uptake of MTX, a key step in its metabolism and clearance, directly influencing its pharmacokinetics and therapeutic efficacy [10]. In contrast, while *SLCO1B3* has overlapping substrate specificity with *SLCO1B1*, it is expressed at lower levels in the liver, potentially limiting its impact on MTX transport. These differences show that *SLCO1B1* plays a more dominant and clinically relevant role in regulating MTX response

The association between haplotype variations in *SLCO1B3* and *SLCO1B1* and the differential response to MTX treatment observed in this study provides valuable insights into the genetic factors influencing drug efficacy in patients with RA. The results revealed distinct haplotype frequency patterns between good and poor responders to MTX therapy, indicating that these polymorphisms can serve as genetic biomarkers for predicting treatment outcomes.

The significant predominance of the GAAT haplotype in good responders, compared with poor responders, indicated a possible protective or facilitative role in MTX metabolism or transport. As *SLCO1B1* and *SLCO1B3* are involved in MTX uptake, the GAAT haplotype may enhance MTX uptake efficiency into hepatocytes, improving drug availability and response [6]. Conversely, the frequency of the TGGT haplotype observed in poor responders revealed an association with reduced drug uptake or increased drug elimination, leading to suboptimal therapeutic MTX levels.

*SLCO1B3* and *SLCO1B1* SNPs may modify hepatocytes, increasing MTX influx and safeguarding the liver [36]. Few studies have investigated the association of *SLCO1B3* and *SLCO1B1* SNPs with MTX-induced liver toxicity in patients with RA. Notably, the exclusive presence of the GGGT haplotype in the good responder group and the TGAC and GGAT haplotypes in the poor responder group may further highlight the complexity of genetic influence on MTX pharmacokinetics. These haplotypes may be critical in determining individual variations in drug pharmacokinetics and should be explored in future studies for their mechanistic roles in MTX metabolism [37].

Although this study highlights the potential of using *SLCO* gene haplotypes as predictive markers for MTX responsiveness, their clinical application requires validation in larger and more diverse cohorts. Furthermore, because the genetic predisposition to drug response is multifactorial and involves numerous genes and environmental factors, comprehensive models that integrate these variables are essential for enhancing predictive accuracy.

In conclusion, this study provided substantial insights into the multifactorial nature of treatment response to MTX in patients with RA, highlighting the lack of association between demographic factors such as age and sex, and the efficacy of MTX, emphasizing the complexity of predicting treatment outcomes based solely on these parameters. However, disease duration emerged as a significant predictor, with longer disease duration correlating with poorer responses to MTX, possibly due to increased joint damage and disease progression. Moreover, this study highlighted the pivotal role of played by genetic factors in determining MTX efficacy and safety, indicating that pharmacogenetic testing could enhance personalized RA treatment strategies.

Investigating the role of *SLCO1B3* and *SLCO1B1* polymorphisms in modulating MTX response in patients with RA provides valuable insights into pharmacogenetics and personalized medicine. Our study findings highlighted the complexity of genetic influences on drug responses, particularly the absence of significant associations between specific *SLCO1B3* polymorphisms and MTX efficacy and safety treatment in RA. In addition, the borderline significant association of *SLCO1B3* polymorphisms with anemia indicates a nuanced role in MTX-related hematological toxicity, thus requiring further investigation.

Although significant findings for *SLCO1B3* were limited, analyzing *SLCO1B1* polymorphisms deepens our understanding of transporter gene variations in collectively influencing MTX pharmacokinetics and pharmacodynamics. These findings emphasize the importance of a personalized approach in RA management, incorporating clinical, demographic, and genetic factors. Such tailored strategies based on genetic profiles hold significant promise for optimizing MTX efficacy, minimizing adverse effects, and improving patient outcomes. Thus, this study paves the way for further research on genetic markers and their practical applications in clinical settings, potentially revolutionizing RA treatment paradigms.

This study has several limitations. The sample size was relatively small, and patient numbers were limited, especially for haplotype analysis. The genetic diversity of human populations introduces complexities in genetic studies, emphasizing the need for research across ethnic groups to broaden applicability. Phenotype complexity also complicated genetic studies, as many traits and diseases involve interactions between multiple genes and environmental factors. Determining the specific genetic variants responsible for a particular trait or disease can be challenging owing to the complex nature of gene–gene and gene–environment interactions. However, even when genetic variants have been identified, their functional implications were not fully understood. Determining the contribution of specific genetic variations to the development of a trait or disease at the molecular level is challenging.

The lack of significant associations in this study does not detract from the potential utility of genetic markers for predicting treatment outcomes; however, it illustrates the intricate interplay of multiple genes and environmental factors affecting drug metabolism and efficacy. Future research should focus on larger, more diverse cohorts to elucidate subtle genetic effects and incorporate advanced genetic analysis techniques such as haplotype and multigene interaction studies. Genome-wide association studies may help identify new genetic markers affecting pharmacokinetics and pharmacodynamics. Adopting a comprehensive approach in future studies could enhance the ability to tailor treatments for individual patients, potentially improving therapeutic outcomes and minimizing adverse effects.

## 4. Materials and Methods

### 4.1. Study Design

This study was part of a project that aims to assess the effect of the different genetic polymorphisms on MTX in patients with RA. The study was a multicenter, hospital-based, case-control study conducted at rheumatology clinics during the period from March 2023 to October 2023. Demographic and clinical data were collected from patient medical records using a designed electronic collecting data sheet. Blood samples from the patients were used for genotyping of *SLCO1B3* and *SLCO1B1* single nucleotide polymorphisms (SNPs).

### 4.2. Ethical Approval and Ethical Consideration

The study protocol was approved by the Unit of Biomedical Research Ethics Committees in the Faculty of Medicine at King Abdulaziz University (Reference No 65-23; on 8 March 2023) and Armed Forces Hospital at Al-Hada (Reference No 2022-692; on 12 December 2022). Informed consent was obtained from all participants prior to collecting their data or samples. To ensure patient privacy and security, each participant was assigned a special code instead of using their names. Participants were free to withdraw from the study at any time.

### 4.3. Subjects and Data Source

The study population consisted of 100 patients with RA, Saudi citizens in Jeddah and Taif. Patients with RA were recruited as outpatients from the clinics of King Abdulaziz University Hospital in Jeddah and Armed Forces Hospital at Al-Hada in Taif city.

#### 4.3.1. Participant Recruitment

Initial screening was conducted on a total of 124 patients (Figure 3). Based on the inclusion and exclusion criteria, 109 patients were found to be eligible for participation in this study. The eligible participants were given the informed consent form, and the primary investigator explained the purpose and process of the research to them. A co-investigator was present as a witness during this process. Out of the eligible patients, nine individuals declined to proceed with the study. Thus, a total of 100 patients agreed to participate and provided complete data for the study.

#### 4.3.2. Inclusion and Exclusion Criteria

During patient clinic visits, a rheumatologist consultant reviewed the medical history of all patients. The assessment included factors such as age, age at RA onset, gender, past medical history, and medication history. Subsequently, the rheumatologist consultant evaluated individuals who met the inclusion criteria. To be included in the study, patients had to be diagnosed with RA and be between the ages of 18 and 80, with a minimum of 6 months of MTX treatment. Conversely, a patient was excluded from the study if they were receiving MTX for a different diagnosis, had chronic liver disease, were unable to attend all study visits or comply with study procedures, or refused to give informed written consent.

### 4.4. Outcomes Measure

A specially designed electronic data collection sheet was developed to gather demographic and clinical data from the 100 patients. The demographic data included age and gender. The clinical data consisted of the patients’ age at RA onset, duration and dose of MTX, whether MTX was administered as monotherapy or in combination with steroid(s), and the Disease Activity Score 28 (DAS-28), which is an index used to assess disease activity and calculated based on the number of tender and swollen joints. Disease state was assessed using the DAS-28 as follows: remission, DAS-28 ≤ 2.6; low disease activity, 2.6 < DAS-28 ≤ 3.2; moderate disease activity, 3.2 < DAS-28 ≤ 5.1; and high disease activity, DAS-28 > 5.1 [19]. Common MTX adverse drug reactions, including gastrointestinal (GI) disturbances, LFT disturbance, and anemia were also documented. In addition, laboratory data such as ESR, CRP, RF, and anti-CCP were collected. Based on their response to MTX, patients were categorized as either good responders or poor responders.

A good drug response (Controls) was defined as a patient receiving a stable dose of MTX for at least 6 months with an ESR < 20 or CPR within the normal range. Conversely, a poor drug response (cases) was defined as the failure to achieve at least a 20% reduction in ESR/CPR despite a minimum 6-month therapy with a dose of at least 15 mg/week [38].

At the end of the visit, a skilled phlebotomist collected 5 mL of whole venous blood samples from the patients for gene analysis.

### 4.5. Selection of SNPs

#### 4.5.1. Candidate Genes and SNP Selection

The *SLCO1B3* and *SLCO1B1* genes were selected for investigation in this study. Based on an extensive literature review, the genetic variants that have been most studied in *SLCO1B3* are (rs4149117) on exon 4 and (rs7311358) on exon 7. Similarly, in the case of *SLCO1B1*, the genetic variants (rs2306283) on exon 5 and (rs4149056) on exon 5 have been widely studied. These variants were chosen due to their known associations with different expressions and functions of the respective genes [39].

#### 4.5.2. Target-Specific Primer Design

The Ensembl genome browser 109 (http://www.ensembl.org/index.html, accessed on 17 July 2025) was utilized to retrieve the whole genomic sequence of *SLCO1B3* and *SLCO1B1.* Subsequently, the start and stop codons of each exon were mapped and labeled. To detect the target locations of genetic variations within the mRNA or protein sequences, the Ensembl genome browser was employed. These variations were then mapped accordingly.

To design the primers, the flanking sequences surrounding the targeted coding regions were inputted into Primer 3, an online application that designs forward and reverse primers. Adjustments were made to the default primer properties, specifically setting the maximum primer size to 22 bases and the maximum primer melting point to 60 ˚C. The selected forward and reverse primer sequences used in this study are listed in Table 12.

### 4.6. Genotyping

#### 4.6.1. DNA Extraction and Quantification

Five milliliters of blood samples were collected from all participants into sterile EDTA tubes for DNA analysis. Genomic DNA was extracted using the QIAamp DNA Mini Kit (Qiagen, Germantown, MD, USA) following the standard protocol. The DNA concentration and purity were assessed using a NanoDrop™ 2000c spectrophotometer (Thermo Scientific, Waltham, MA, USA). Detailed steps for the DNA extraction process have been described previously in Magadmi et al. [40].

#### 4.6.2. Polymerase Chain Reaction (PCR)

To confirm the presence and size of the PCR product, PCR amplification was conducted for all DNA samples prior to sequencing. The detailed PCR protocol, including primer sequences and thermal cycling conditions, has been described in Magadmi et al. [41]. The integrity of PCR products was confirmed using agarose gel electrophoresis (Edvotek Inc., Washington, DC, USA) for 40 min at 90 V. The bands were visualized under UV light using the ChemiDoc-It Imaging System (Analytik Jena, Jena, Germany).

#### 4.6.3. Sanger Sequencing

The amplified PCR products were subjected to Sanger sequencing using the BigDye Terminator v3.1 Cycle Sequencing Kit (Thermo Fisher Scientific, Waltham, MA, USA). Sequencing reactions were performed in a Veriti 96-Well Fast Thermal Cycler (Applied Biosystems, Waltham, MA, USA). The plates were loaded into the 3500 Genetic Analyzer (Thermo Fisher Scientific, Waltham, MA, USA) following the manufacturer’s recommended settings. Further details of the sequencing procedure are available in Magadmi et al. [41]. Alignment and identifying of the sequence variants were performed using BioEdit software version 6. The sequences were visualized using BioEdit software and further analyzed by National Center for Biotechnology Information (NCBI) blast.

### 4.7. Statistical Analysis

Demographic and clinical data were analyzed using Statistical Package for Social Sciences (SPSS) for Mac version 29.0 software (SPSS Inc., Chicago, IL, USA). Categorical variables of the patients are represented as frequencies and percentages, while continuous variables are represented as mean ± standard deviation (SD). To compare the differences between the “good responders” and “poor responders” groups, the Pearson’s chi-square test or *Fisher’*s test was performed for categorical variables. Allele and genotype frequencies, as well as Hardy–Weinberg equilibrium (HWE), were analyzed using SNPStats [41]. Analysis of association with the response variable was conducted using linear or logistic regression, considering multiple inheritance models such as co-dominant, dominant, and recessive. Age, disease duration, and steroid use were included as covariates in the regression models to adjust for potential confounding factors. In addition, multicollinearity among predictor variables was assessed using the Variance Inflation Factor (VIF) in SPSS, and all VIF values were below five, indicating no significant multicollinearity. The association of haplotypes with the response was also examined. Statistical significance was set at *p <* 0.05. Furthermore, 95% confidence intervals (CIs) and odds ratios (ORs) were calculated.

## Figures and Tables

**Figure 1 pharmaceuticals-18-01069-f001:**
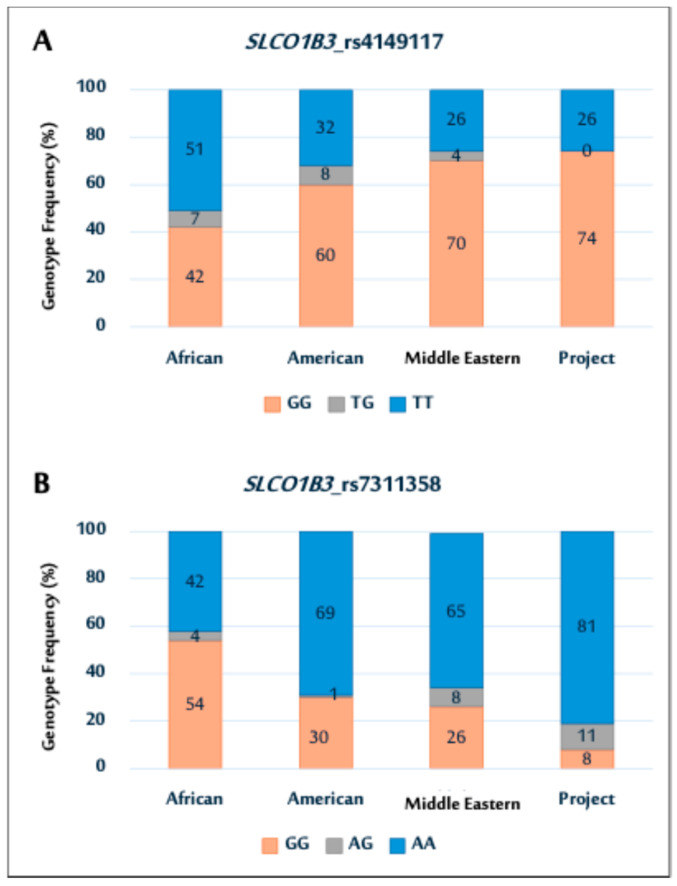
Genotype distributions of *SLCO1B3* polymorphisms (rs4149117 (**A**); and rs7311358 (**B**)) among African, American, Middle Eastern populations, and current Saudi cohort.

**Figure 2 pharmaceuticals-18-01069-f002:**
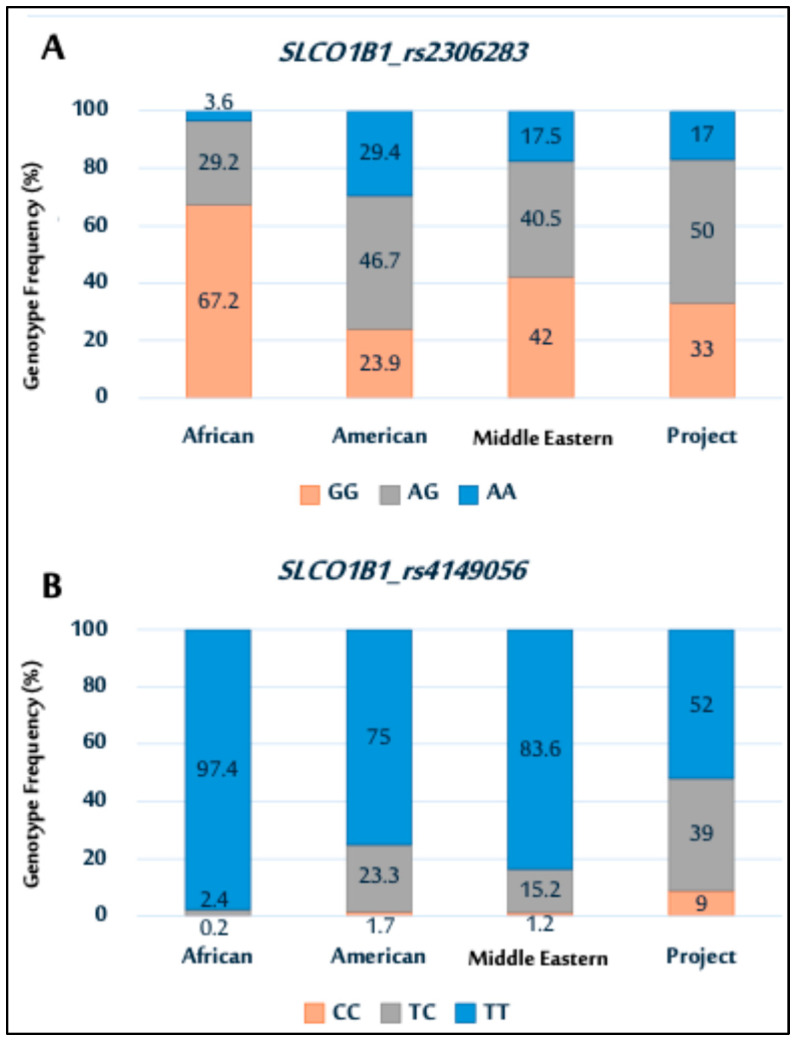
Genotype distributions of *SLCO1B1* polymorphisms (rs2306283 (**A**); and rs4149056 (**B**)) among African, American, Middle Eastern populations, and current Saudi cohort.

**Figure 3 pharmaceuticals-18-01069-f003:**
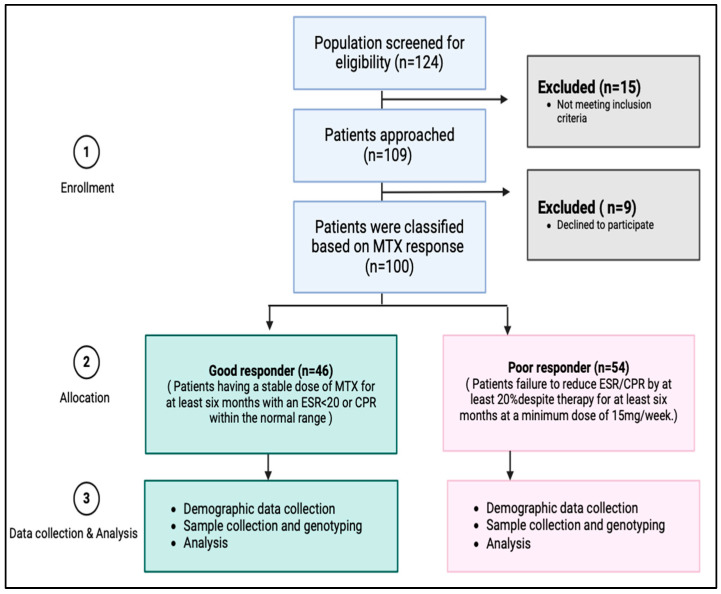
Study flow chart of subjects’ enrollment and analysis.

**Table 1 pharmaceuticals-18-01069-t001:** Demographic and clinical characteristics of patients (N = 100).

Feature		Mean	SD
Age (years)		53.02	12.78
Age of onset (years)		46.26	12.35
		**Frequency**	**Percentage **
Disease duration	6 months–1 year	13	13%
	>1 year	87	87%
Sex	Male	20	20%
	Female	80	80%
**MTX**		**Mean**	** SD **
MTX dose		14.98	3.7
**MTX**		**Frequency**	** Percentage **
Duration of MTX treatment	6 months–1 year	15	15%
	>1 year	85	85%
Regimen type	Monotherapy	44	44%
	Combined therapy	56	56%
** Markers of disease activity **		**Mean**	** SD **
DAS-28		2.94	0.96
CRP		10.56	7.4
ESR		27.44	17.33
** Markers of disease activity **		**Frequency**	** Percentage **
RF	+ve	68	68%
	−ve	28	28%
	Not known	4	4%
Anti-CCP	+ve	62	62%
	−ve	15	15%
	Not known	23	23%
**Adverse drug reaction**		**Frequency**	** Percentage **
Adverse drug reaction ^§^	GI disturbance	61	61%
	Increased liver enzymes	25	25%
	Anemia	14	14%
**Classification of RA based on response to MTX**		**Frequency**	** Percentage **
Patient response	Good responders	46	46%
	Poor responders	54	54%

Anti-CCP: anti-cyclic citrullinated peptide; CRP: C-reactive protein; DAS-28: disease activity score 28; ESR: erythrocyte sedimentation rate; MTX: methotrexate; RA: rheumatoid arthritis; RF: rheumatoid factor; SD: standard deviation. ^§^ A patient may have more than one adverse drug reaction.

**Table 2 pharmaceuticals-18-01069-t002:** Association of demographic and clinical data with patients’ response to methotrexate (N = 100).

Feature		Good Responders (N = 46)	Poor Responders (N = 54)	*p*-Value
Age (years)		52.50 (12.64)	53.416 (13.00)	0.709 *^t^*
Age of onset (years)		47.79 (11.76)	44.98 (12.80)	0.264 ^*t*^
Disease duration	6 months–1 year	11 (23.9)	2 (3.7)	0.009 *
	>1 year	35 (76.1)	52 (96.3)	
Sex	Male	8 (17.4)	12 (22.2)	0.621
	Female	38 (82.6)	42 (77.8)	
**MTX**				
MTX dose (mean)		15.05 (3.00)	14.91 (4.23)	0.844 ^*t*^
Duration of MTX treatment	6 months–1 year	10 (21.7)	5 (9.3)	0.098 *^F^*
	>1 year	36 (78.3)	49 (90.7)	
Regimen type	Monotherapy	30 (65.2)	14 (25.9)	<0.001 *
	Combined therapy (with steroid)	16 (34.6)	40 (74.1)	
** Markers of disease activity **				
DAS-28 (mean (SD))		2.36 (0.90)	3.43 (0.70)	<0.001 **^t^*
CRP (mean (SD))		7.74 (4.41)	12.96 (8.54)	<0.001 **^t^*
ESR (mean (SD))		23.70 (15.86)	30.63 (18.02)	0.046 **^t^*
RF	+ve	32 (69.6)	36 (66.7)	0.690
	−ve	13 (28.3)	15 (27.8)	
	Not known	1 (2.2)	3 (5.6)	
Anti-CCP	+ve	30 (65.2)	32 (59.3)	0.442
	−ve	8 (17.4)	7 (13.0)	
	Not known	8 (17.4)	15 (27.8)	
**Adverse drug reaction**				
Adverse drug reaction ^§^	GI disturbance	30 (30)	31 (31)	0.642
	Increased liver enzymes	11 (11)	14 (14)	
	Anemia	5 (5)	9 (9)	

Anti-CCP: anti-cyclic citrullinated peptide; CRP: C-reactive protein; DAS-28: disease activity score 28; ESR: erythrocyte sedimentation rate; MTX: methotrexate; RA: rheumatoid arthritis; RF: rheumatoid factor; SD: standard deviation. ^§^ A patient may have more than one adverse drug reaction. * Statistically significant difference (*p <* 0.05). Test used: *^t^* independent *t*-test; *^F^* Fisher’s exact test: otherwise, *chi-square* test was used.

**Table 3 pharmaceuticals-18-01069-t003:** Genotype distributions and allele frequencies of *SLCO1B3* polymorphisms among all patients.

Genotype/Allele	N (%)
** *SLCO1B3* ** **_rs4149117 (N = 100)**	
GG	74 (74)
TG	0 (0)
TT	26 (26)
G	148 (74)
T	52 (26)
** *SLCO1B3* ** **_rs7311358 (N = 100)**	
AA	81(81)
AG	11 (11)
GG	8 (8)
A	173 (87)
G	27 (13)

Data are presented as number of patients (N) and percentages (%).

**Table 4 pharmaceuticals-18-01069-t004:** Genotype and allele frequencies of *SLCO1B3* gene polymorphisms among good and poor responders to methotrexate.

Genotype/Allele	Good Responders N (%)	Poor Responders N (%)	Adjusted OR (95% CI)	*p*-Value
	N = 46	N = 54		
***SLCO1B3*_rs4149117 (N = 100)**				
GG	35 (76.1%)	39 (72.2%)	0.81 (0.33–2.013)	0.66
TT	11 (23.9%)	15 (27.8%)	1.2 (0.496–3.01)	
G	70 (76.1%)	78 (72.2%)	1.2 (0.64–2.31)	0.53
T	22 (23.9%)	30 (27.8%)	0.81 (0.43–1.54)	
***SLCO1B3*_rs7311358 (N = 100)**				
AA	39 (84.8%)	42 (77.8%)	1.5 (0.56–4.45)	0.45
AG	5 (10.9%)	6 (11.1%)	0.97 (0.27–3.4)	
GG	2 (4.3%)	6 (11.1%)	0.36 (0.06–1.8)	
A	83 (90%)	90 (83%)	1.84 (0.78–4.33)	0.16
G	9 (10%)	18 (17%)	0.54 (0.23–1.27)	

Data are presented as number of patients (N) and percentages (%). Data were analyzed using chi-square test. OR: odds ratio; CI: confidence interval. The OR was estimated using logistic regression analysis after adjusting for the regimen.

**Table 5 pharmaceuticals-18-01069-t005:** Haplotype frequencies of *SLCO1B3* polymorphisms among good and poor responders to methotrexate (N = 100).

Haplotype		Haplotype Frequency		Adjusted OR (95% CI)	*p*-Value
*SLCO1B3* _rs4149117	*SLCO1B3* _rs7311358	Good Responder (N = 46)	Poor Responder (N = 54)		
G	A	70	78	1	-
T	A	14	13	0.78 (0.41–1.51)	0.47
T	G	6	15	1.76 (0.78–4.01)	0.18
G	G	2	2	0.95 (0.23–3.86)	0.94

Data are presented as the number of patients (N) and were analyzed using logistic regression.

**Table 6 pharmaceuticals-18-01069-t006:** Association between genotype distributions of *SLCO1B3* gene polymorphisms and the risk of GI disturbance, increased liver enzymes, and anemia induced by methotrexate among patients with rheumatoid arthritis.

Genotype	GI Disturbance (N = 61)	*p*-Value	Increased Liver Enzymes (N = 25)	*p*-Value	Anemia (N = 14)	*p*-Value
	*SLCO1B3*_rs4149117					
TT	17 (27.9)	0.25	6 (24)	0.549	3 (21.4)	0.0463 *
GG	44 (72.1)		19 (76)		11 (78.6)	
** *SLCO1B3* ** **_rs7311358**						
AA	48 (78.7)	0.084	20 (80)	0.576	13 (92.9)	0.0487 *
AG	10 (16.4)		1 (4)		0	
GG	3 (4.9)		4 (16)		1 (7.1)	

Data are presented as number of patients (N) and percentages (%). * Statistically significant difference (*p <* 0.05). Data were analyzed using the chi-square test.

**Table 7 pharmaceuticals-18-01069-t007:** Genotype distributions and allele frequencies of *SLCO1B1* polymorphisms among all patients.

Genotype/Allele	N (%)
** *SLCO1B1* ** **_rs2306283 (N = 100)**	
AA	17 (17)
AG	50 (50)
GG	33 (33)
A	84 (42)
G	116 (58)
** *SLCO1B1* ** **_rs4149056 (N = 100)**	
TT	52 (52)
TC	39 (39)
CC	9 (9)
T	143 (72)
C	57 (28)

Data are presented as number of patients (N) and percentages (%).

**Table 8 pharmaceuticals-18-01069-t008:** Genotype and allele frequencies of *SLCO1B1* gene polymorphisms among good and poor responders to methotrexate.

Genotype/Allele	Good Responders N (%)	Poor Responders N (%)	Adjusted OR (95% CI)	*p*-Value
	N = 46	N = 54		
** *SLCO1B1* ** **_rs2306283 (N = 100)**				
AA	8 (17.4%)	9 (16.7%)	1.05 (0.36–2.9)	0.18
AG	27 (58.7%)	23 (42.6%)	1.9 (0.89–4.3)	
GG	11 (23.9%)	22 (40.7%)	0.46 (0.195–1.08)	
A	43 (47%)	41(38%)	1.43 (0.81–2.52)	0.21
G	49 (53%)	67 (62%)	0.69 (0.39–1.22)	
** *SLCO1B1* ** **_rs4149056 (N = 100)**				
TT	27 (58.7%)	25 (46.3%)	1.6 (0.74–3.6)	0.44
TC	15 (32.6%)	24 (44.4%)	0.6 (0.26–1.3)	
CC	4 (8.7%)	5 (9.3%)	0.93 (0.23–3.7)	
T	69 (75%)	74 (69%)	1.37 (0.73–2.5)	0.31
C	23 (25%)	34 (31%)	0.72 (0.38–1.35)	

Data are presented as number of patients (N) and percentages (%). Data were analyzed using the chi-square test. OR: odds ratio; CI: confidence interval. The OR was estimated via logistic regression analysis after adjusting for the regimen.

**Table 9 pharmaceuticals-18-01069-t009:** Haplotype frequencies of *SLCO1B1* polymorphisms among good and poor responders to methotrexate (N = 100).

Haplotype		Haplotype Frequency		Adjusted OR (95% CI)	*p*-Value
*SLCO1B1* _rs2306283	*SLCO1B1* _rs4149056	Good Responder (N = 46)	Poor Responder (N = 54)		
A	T	45	36	0.46 (0.19–1.09)	0.01 *
G	T	28	37	1.24 (0.61–2.50)	0.55
G	C	15	32	1.71 (0.81–3.62)	0.16
A	C	4	3	0.86 (0.20–3.63)	0.84

Data are presented as number of patients (N). *: statistically significant difference (*p <* 0.05). Data were analyzed using logistic regression.

**Table 10 pharmaceuticals-18-01069-t010:** Association between genotype distributions of *SLCO1B1* gene polymorphisms and the risk of methotrexate-induced adverse reactions.

Genotype	GI Disturbance (N = 61)	*p*-Value	Increased Liver Enzymes (N = 25)	*p*-Value	Anemia (N = 14)	*p*-Value
	*SLCO1B1*_ rs2306283					
AA	7 (11.5)	0.001 *	5 (20)	0.111	5 (37.7)	0.009 *
AG	25 (40.9)		16 (64)		9 (64.3)	
GG	29 (47.6)		4 (16)		0	
** *SLCO1B1* ** **_rs4149056**						
TT	24 (39.3)	0.007 *	19 (76)	0.021 *	9 (64.3)	0.61
TC	30 (49.2)		5 (20)		4 (28.6)	
CC	7 (11.5)		1 (4)		1 (7.1)	

Data are presented as number of patients (N) and percentages (%). *: Statistically significant difference (*p <* 0.05). Data were analyzed using the chi-square test.

**Table 11 pharmaceuticals-18-01069-t011:** Haplotype frequencies of *SLCO1B3* and *SLCO1B1* polymorphisms among good and poor responders to MTX (N = 100).

Haplotype				Haplotype Frequency		*p*-Value
*SLCO1B3* rs4149117	*SLCO1B3* rs7311358	*SLCO1B1* rs2306283	*SLCO1B1* rs4149056	Good Responder (N = 46)	Poor Responder (N = 54)	
G	A	A	T	33	28	0.01 *
G	A	G	T	18	20	0.2
G	A	G	C	16	19	0.2
T	A	G	T	7	7	0.18
T	G	G	T	4	7	0.01 *
T	A	A	T	7	1	0.2
T	G	G	C	1	4	0.2
G	A	A	C	4	2	0.18
T	A	G	C	1	2	0.2
T	G	A	T	1	1	0.2
G	G	A	T	0	1	0.18
G	G	G	T	2	0	0.01 *
T	G	A	C	0	1	0.2

Data are presented as the number of patients (N). Data were analyzed using logistic regression analysis. * Statistically significant difference (*p <* 0.05).

**Table 12 pharmaceuticals-18-01069-t012:** Forward and reverse primer sequences of selected SNPs.

SNP ID	Exon Location	Amino Acid Change Location	Protein Change Location	Forward Primer	Reverse Primer
rs4149117	Exon 4	c.334T>G	p.Ser112Ala	CTACACAGACCGAAGTTA	ATGGGAACTGGAAGTATT
rs7311358	Exon 7	699G>A	p.Met235Ile	GTAGTTTGAATGCAATAG	GCACTGGGATCTCTGTTT
rs2306283	Exon 5	388A>G	p.Asn130Asp	GCATCACCTGAGATAGTA	ACAGGTATTCTAAAGAAT
rs4149056	Exon 5	521T>C	p.V174A	TCGTGGAATAGGGGAGAC	ACATGTGGATATATGTGT

## Data Availability

Data is contained within the article.

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
