# Peer review of "Association of SLCO1B3 and SLCO1B1 Polymorphisms with Methotrexate Efficacy and Toxicity in Saudi Rheumatoid Arthritis Patients"

_pharmaceuticals, 2025, doi:10.3390/ph18071069_

Round 1
Reviewer 1 Report
Comments and Suggestions for Authors
Comments on the manuscript titled “Association of SLCO1B3 and SLCO1B1 Polymorphisms with Methotrexate Efficacy and Toxicity among Saudi Patients with Rheumatoid Arthritis”
Overall, I find this manuscript to be a well-written and valuable contribution to the field, particularly because it addresses a significant knowledge gap regarding the pharmacogenetics of methotrexate (MTX) in the Saudi Arabian population. Given the scarcity of genetic data on SLCO1B3 and SLCO1B1 polymorphisms in this demographic, the study provides important insights that could inform personalized treatment strategies for rheumatoid arthritis (RA) patients in the region.
However, I have several suggestions that could enhance the clarity, rigor, and impact of the paper:
- Gene Nomenclature Formatting
Please ensure that all gene names (e.g., SLCO1B3, SLCO1B1) are consistently italicized throughout the manuscript in accordance with standard genetic nomenclature conventions. This will improve the professionalism and readability of the text. - Abstract Editing
The abstract contains a repeated phrase—“remains limited”—within the same sentence, which appears to be an editorial oversight. I recommend revising this sentence to avoid redundancy and improve clarity. - Rationale for SNP Selection
The manuscript would benefit from a more detailed explanation regarding the selection of the four single nucleotide polymorphisms (SNPs): SLCO1B3 (rs4149117, rs7311358) and SLCO1B1 (rs2306283, rs4149056). Clarifying why these particular variants were chosen—whether due to prior associations with MTX pharmacokinetics, functional relevance, or allele frequency in the population—would strengthen the study’s rationale.
Moreover, incorporating bioinformatic analyses using publicly available resources such as the UALCAN or GTEx portals could provide supportive evidence on the expression profiles and functional impact of these SNPs, thereby reinforcing the justification for their investigation. - Comparative Population Analysis
It would be highly informative if the authors could compare the allele frequencies and haplotype distributions of these SNPs in the Saudi population with those from other continental populations (e.g., European, East Asian, African). Such comparative analysis could highlight population-specific genetic differences and potentially explain variability in MTX response across ethnic groups. - Correlation with Disease Epidemiology
Exploring associations between the studied SNPs and the incidence or prevalence of RA within the Saudi Arabian population could add an epidemiological dimension to the genetic findings. This might help elucidate whether these polymorphisms contribute not only to drug response but also to disease susceptibility or severity in this specific population. - Improvement of Figures
The quality of Figures 1 and 2 could be enhanced by using higher resolution images to improve visual clarity and interpretability. The authors might consider employing advanced graphical tools or software such as GraphPad Prism or R (with packages like ggplot2) to generate more polished and publication-quality figures. Clear and visually appealing figures will greatly aid readers in understanding the key results.
Author Response
Comments 1: Gene Nomenclature Formatting |
Response 1: We have carefully reviewed the manuscript and ensured that all gene names are italicized consistently.
|
Comments 2: Abstract Editing: The abstract contains a repeated phrase—“remains limited”—within the same sentence, which appears to be an editorial oversight. I recommend revising this sentence to avoid redundancy and improve clarity. |
Response 2: Thank you for your comment. The abstract has been revised to remove the redundancy and improve clarity.
|
Comments 3: Rationale for SNP Selection: The manuscript would benefit from a more detailed explanation regarding the selection of the four single nucleotide polymorphisms (SNPs): SLCO1B3 (rs4149117, rs7311358) and SLCO1B1 (rs2306283, rs4149056). Clarifying why these particular variants were chosen—whether due to prior associations with MTX pharmacokinetics, functional relevance, or allele frequency in the population—would strengthen the study’s rationale. Moreover, incorporating bioinformatic analyses using publicly available resources such as the UALCAN or GTEx portals could provide supportive evidence on the expression profiles and functional impact of these SNPs, thereby reinforcing the justification for their investigation. |
Response 3: We have expanded the rationale for selecting the four SNPs (SLCO1B3: rs4149117, rs7311358; SLCO1B1: rs2306283, rs4149056). These variants were chosen due to their prior associations with methotrexate (MTX) pharmacokinetics and their functional relevance in drug transport mechanisms (Page 22, Section 4.5.1, lines 731-737)..
The authors thank the reviewer for their valuable suggestion regarding the use of bioinformatics analyses. We would like to share that the research group is currently conducting a related project that utilizes bioinformatics tools and incorporates additional genes. This project is now in the advanced stages, with the manuscript preparation process already underway.
|
Comments 4: Comparative Population Analysis: It would be highly informative if the authors could compare the allele frequencies and haplotype distributions of these SNPs in the Saudi population with those from other continental populations (e.g., European, East Asian, African). Such comparative analysis could highlight population-specific genetic differences and potentially explain variability in MTX response across ethnic groups. |
Response 4: We thank the reviewer for their insightful suggestion to compare the allele frequencies and haplotype distributions of the studied SNPs in the Saudi population with those from other continental populations. In our study, we compared the genotype distributions of the studied SNPs (SLCO1B3 rs4149117 and rs7311358, and SLCO1B1 rs2306283 and rs4149056) with those from other populations, including Middle Eastern, African, East Asian, and European populations, using publicly available resources such as the National Center for Biotechnology Information (NCBI) (Figure 1 and 2). These comparisons provided indirect estimates of allele frequencies, highlighting notable ethnic variations. We agree that a more detailed comparative analysis of haplotype distributions would further enhance the study. However, due to the limited sample size and the scope of this study, a comprehensive haplotype analysis across populations was not performed at this stage.
|
Comments 5: Correlation with Disease Epidemiology: Exploring associations between the studied SNPs and the incidence or prevalence of RA within the Saudi Arabian population could add an epidemiological dimension to the genetic findings. This might help elucidate whether these polymorphisms contribute not only to drug response but also to disease susceptibility or severity in this specific population. |
Response 5: We thank the reviewer for their thoughtful suggestion to explore the associations between the studied SNPs and the incidence or prevalence of rheumatoid arthritis (RA) within the Saudi population. While we agree that such an analysis could provide valuable insights into the potential role of these SNPs in disease susceptibility or severity, this topic extends beyond the scope of our current study. Our research specifically focuses on the pharmacogenetics of methotrexate (MTX), with an emphasis on how genetic polymorphisms in SLCO1B3 and SLCO1B1 influence the pharmacokinetics of MTX, ultimately affecting drug response and safety in patients with RA. The primary objective of our study was to investigate the association of SLCO1B3 and SLCO1B1 polymorphisms with MTX efficacy and toxicity, rather than exploring their role in the pathogenesis or epidemiology of RA. Notably, existing research has already provided substantial insights into the role of genetic factors in RA pathogenesis, such as studies on HLA alleles and other candidate genes.
We thank the reviewer for their valuable input, which highlights an important avenue for future research. Building on our findings, future studies could explore the broader interplay between genetic polymorphisms, disease susceptibility, and drug response to further advance personalized medicine for RA management.
|
Comments 6: Improvement of FiguresThe quality of Figures 1 and 2 could be enhanced by using higher resolution images to improve visual clarity and interpretability. The authors might consider employing advanced graphical tools or software such as GraphPad Prism or R (with packages like ggplot2) to generate more polished and publication-quality figures. Clear and visually appealing figures will greatly aid readers in understanding the key results.. |
Response 6: We thank the reviewer for their constructive comment regarding the improvement of Figures 1 and 2. As per the suggestion, we have re-generated the figures using high-quality images to enhance visual clarity and interpretability. Again, we highly appreciate the reviewer’s valuable feedback in helping us improve the quality and presentation of our work. |

Reviewer 2 Report
Comments and Suggestions for Authors
This manuscript appears to be a well-structured and comprehensive pharmacogenetic study investigating the association between SLCO1B3 and SLCO1B1 polymorphisms and methotrexate (MTX) efficacy and toxicity in Saudi patients with rheumatoid arthritis (RA).
- The current title is lengthy. Consider simplifying:
Association of SLCO1B3 and SLCO1B1 Polymorphisms with Methotrexate Efficacy and Toxicity in Saudi Rheumatoid Arthritis Patients
- The abstract could briefly mention the sample size (N=100) and key statistical methods (e.g., logistic regression, haplotype analysis).
- Specify which SNPs were studied (rs4149117, rs7311358, rs2306283, rs4149056).
- Table 1 (Demographics): Consider adding a column for p-values if comparing baseline characteristics between responders/non-responders.
- Figures 1 & 2 (Genotype Distributions): Ensure axis labels are clear (e.g., Genotype Frequency (%).
- Add a brief legend explaining population comparisons.
- Briefly discuss how SLCO1B1/1B3 variants may alter MTX transport (e.g., reduced hepatic uptake leading to toxicity).
- Suggest how findings could be applied (e.g., Pre-treatment genotyping may identify high-risk patients for closer monitoring).
- Acknowledge potential confounding factors (e.g., concomitant medications, diet).
- Suggest future directions (e.g., larger multi-ethnic studies, functional validation).
- Ensure all cited studies (e.g., Jenko et al., Banach et al.) are correctly referenced.
- Include recent meta-analyses on MTX pharmacogenetics (if available).
Author Response
1. Summary |
|
|
Thank you very much for taking the time to review this manuscript. Please find the detailed responses below and the corresponding revisions highlighted in green in the re-submitted files.
|
||
3. Point-by-point response to Comments and Suggestions for Authors |
||
Comments 1: The current title is lengthy. Consider simplifying: Association of SLCO1B3 and SLCO1B1 Polymorphisms with Methotrexate Efficacy and Toxicity in Saudi Rheumatoid Arthritis Patients. |
||
Response 1: The authors deeply appreciate the reviewer’s suggestion and agree that the title can be simplified for clarity and conciseness. As suggested, we have revised the title to: "Association of SLCO1B3 and SLCO1B1 Polymorphisms with Methotrexate Efficacy and Toxicity in Saudi Rheumatoid Arthritis Patients." Highlighted in green.
|
||
Comments 2: The abstract could briefly mention the sample size (N=100) and key statistical methods (e.g., logistic regression, haplotype analysis). Comments 3: Specify which SNPs were studied (rs4149117, rs7311358, rs2306283, rs4149056). |
||
Response 2 & 3: Thank you for your comment. We have revised the abstract to include the sample size (N=100) and the key statistical methods, including logistic regression and haplotype analysis. . (Highlighted in green.). The specific SNPs studied (rs4149117, rs7311358, rs2306283, rs4149056) are now explicitly mentioned in the abstract. (Highlighted in green.).
|
||
Comments 4: Table 1 (Demographics): Consider adding a column for p-values if comparing baseline characteristics between responders/non-responders. |
||
Response 4: As suggested, we have clarified that Table 3 includes the p-values comparing the baseline characteristics between responders and non-responders.
|
||
Comments 5: Figures 1 & 2 (Genotype Distributions): Ensure axis labels are clear (e.g., Genotype Frequency (%). Comments 6: Add a brief legend explaining population comparisons. |
||
Response 5 & 6: We thank the reviewer for their insightful suggestion. The axis labels in Figures 1 and 2 have been updated to clearly indicate Genotype Frequency (%). A brief legend has been added to explain the population comparisons illustrated in the figures. . (Highlighted in green.). |
||
Comments 7: Briefly discuss how SLCO1B1/1B3 variants may alter MTX transport (e.g., reduced hepatic uptake leading to toxicity). |
||
Response 7: We thank the reviewer for their valuable input. We have included a discussion elaborating on how SLCO1B1 and SLCO1B3 variants may alter MTX transport (Page 18-19, Lines: 581-588) . (Highlighted in green.).
|
||
Comments 8: Suggest how findings could be applied (e.g., Pre-treatment genotyping may identify high-risk patients for closer monitoring). |
||
Response 8: We thank the reviewer for their constructive comment. The potential application of our findings has been emphasized, suggesting that pre-treatment genotyping of SLCO1B1 and SLCO1B3 could identify high-risk patients for closer monitoring and personalized treatment strategies to improve MTX efficacy and safety (Page 19-20, Lines: 635-641). (Highlighted in green.).
|
||
Comments 9: Acknowledge potential confounding factors (e.g., concomitant medications, diet). |
||
Response 9: We have acknowledged potential confounding factors, including concomitant medications, diet, and other environmental influences. (Page 19, Lines: 610-615).
|
||
Comments 10: Suggest future directions (e.g., larger multi-ethnic studies, functional validation). |
||
Response 10: We thank the reviewer for their valuable comment. In the revised manuscript, we have highlighted the future directions. (Page 20, Lines: 656 - 661). (Highlighted in green.).
|
||
Comments 11: Ensure all cited studies (e.g., Jenko et al., Banach et al.) are correctly referenced. |
||
Response 11: All cited studies have been verified for accuracy in the references section.
|
